# Comparative Pharmacokinetics of Sulfadiazine and Its Metabolite N4-Acetyl Sulfadiazine in Grass Carp (*Ctenopharyngodon idella*) at Different Temperatures after Oral Administration

**DOI:** 10.3390/pharmaceutics14040712

**Published:** 2022-03-26

**Authors:** Ning Xu, Miao Li, Zhoumeng Lin, Xiaohui Ai

**Affiliations:** 1Yangtze River Fisheries Research Institute, Chinese Academy of Fishery Sciences, No. 8 Wuda Park Road 1, Wuhan 430223, China; xuning@yfi.ac.cn; 2Institute of Computational Comparative Medicine (ICCM), Department of Anatomy and Physiology, College of Veterinary Medicine, Kansas State University, 1800 Denison Avenue, Manhattan, KS 66506, USA; miaoli@ksu.edu; 3Hu Bei Province Engineering and Technology Research Center of Aquatic Product Quality and Safety, 8 Wuda Park Road 1, Wuhan 430223, China; 4Department of Environmental and Global Health, College of Public Health and Health Professions, University of Florida, 1225 Center Drive, Gainesville, FL 32610, USA; 5Center for Environmental and Human Toxicology, University of Florida, 2187 Mowry Road, Gainesville, FL 32608, USA

**Keywords:** temperature, pharmacokinetics, sulfadiazine, N_4_-acetyl sulfadiazine, grass carp

## Abstract

In this study, the plasma pharmacokinetics and tissue disposition of sulfadiazine (SDZ) and its main metabolite, N_4_-acetyl sulfadiazine (ACT-SDZ), were compared between 18 and 24 °C following a single oral administration of SDZ at 50 mg/kg in grass carp (*Ctenopharyngodon idella*). The plasma and tissues were sampled from 0.167 h up to 96 h and analyzed by ultra-performance liquid chromatography with an ultraviolet detector. The pharmacokinetic parameters were estimated using a one-compartmental approach. Results showed that pharmacokinetics of SDZ and ACT-SDZ in plasma and tissues were notably influenced by the increase of temperature. The increased temperature shortened the absorption half-life (K01_HL) of SDZ and ACT-SDZ in gill, kidney, and plasma, but increased in liver and muscle + skin. The elimination half-life (K10_HF) and the area under concentration-time curve (AUC_0–∞_) of SDZ and ACT-SDZ all presented a declined trend. The apparent volume of distribution (V_F) of SDZ in plasma was increased from 0.93 to 1.64 L/kg, and the apparent systemic total body clearance (Cl_F) was also increased from 0.01 to 0.05 L/h/kg. Overall, the rise of temperature decreased K10_HF, AUC_0–∞_ of SDZ, and ACT-SDZ in plasma and tissues, but increased V_F and Cl_F in the plasma for SDZ.

## 1. Introduction

Sulfadiazine (SDZ) is a commonly used synthetic drug that belongs to sulfonamides possessing broad-spectrum antibacterial activity for Gram-negative and Gram-positive bacteria by competitive antagonism of p-aminobenzoic acid to inhibit bacterial DNA synthesis [1]. The use of SDZ in veterinary and human medicine has a long history from the last century when development of this drug began in 1935. Due to its excellent antibacterial activity against *Vibrio alginolyticus*, *Photobacterium damselae spp*, *Aeromonas hydrophilia, Edwardsiella ictalurid,* and *Streptococcus spp* of aquatic animals [2,3,4,5], it has been also approved to cure fish diseases in many countries, such as India, Malaysia, Singapore, China, and the Philippines [6]. Therefore, it is important to investigate the pharmacokinetic characteristics of SDZ in fish in order to design a rational dosage regimen and prevent the occurrence of drug resistance.

Grass carp (*Ctenopharyngodon idella*) is the top cultured fish species in global freshwater aquaculture, with an estimated global production of more than 6.07 million tons per year with the majority of the production in Asia [7], in part because of its rapid growth and the broad utilization of feed. To accommodate progressively increasing market demands, the intensive cultured system was introduced in aquaculture to enlarge the production, but the cost of the intensive cultured system is that pathogens are easier to transmit among fish under the confined culturing environments [8,9]. In this case, antibiotics are still an effective method for reducing fish mortality and economic loss [10,11]. Currently, it is reported that grass carp was susceptible to *Aeromonas hydrophila*, *Fibrobacter columnaris*, *Pseudomonas fluorescens*, and *Vibrio vulnificus* [12,13,14,15]. Fortunately, SDZ is still a sensitive drug for these bacterial diseases despite an improved dosage being needed. Consequently, it has imperative significance to study plasma pharmacokinetics and tissue disposition of SDZ in grass carp.

To date, the pharmacokinetic profile of SDZ has been investigated in various fish species, such as mandarin fish (*Siniperca chuatsi*) [16], Gilthead sea bream (*Sparus aurata*) [17], Atlantic salmon (*Salmo salar*) [18], rainbow trout (*Oncorhynchus mukiss*) [19], and channel catfish [20]. For grass carp, our lab recently determined that the oral bioavailability of SDZ at a water temperature of 24.0 °C was 22.34% [21]. However, it is important to note that the fish are heterothermic animals, and the temperature in the living environment has an important effect on the absorption, distribution, metabolism, and excretion of drugs in fish. The important impact of water temperature in drug pharmacokinetics in fish has been demonstrated with many other drugs, including doxycycline [22], enrofloxacin [23], florfenicol (FF) [24,25,26], oxolinic acid [27], flumequine [28] and oxytetracycline [29,30]. However, currently, no information is available on the effect of water temperature on the pharmacokinetics of SDZ in grass carp. Moreover, among existing pharmacokinetic studies in fish, very few have also determined the pharmacokinetic profile of SDZ metabolite, N_4_-acetyl sulfadiazine (ACT-SDZ), which is found as the dominant metabolite in grass carp from preliminary metabolic research in our lab. Therefore, the objective of this study was to examine the pharmacokinetic profiles of SDZ and its metabolite ACT-SDZ in the plasma and tissues in grass carp following a single oral administration at two different temperatures.

## 2. Materials and Methods

### 2.1. Chemicals and Reagents

SDZ and ACT-SDZ with high purity (99%) were obtained from Dr. Ehrenstorfer GmbH. (Augsburg, Germany). Commercial SDZ powder with a purity grade of 98% used for oral gavage was purchased from Zhongbo Aquaculture Biotechnology Co., Ltd. (Wuhan, China). The high-performance liquid chromatography (HPLC) grade solvents including acetonitrile, ethyl acetate, methanol, and water were purchased from Thermo Fisher (Waltham, MA, USA) and J–T Baker (Philipsburg, SX, USA). Sodium hydroxide and anhydrous magnesium were purchased from Shanghai Guoyao Company (Shanghai, China). The 1.5-mL vials and centrifugal tubes provided by Shanghai CNW Technologies (Shanghai, China) were used for instrumental analysis.

### 2.2. Animals and Management

One hundred and fifty grass carp (450.3 ± 58.9 g, 12 months of age, mixed genders) provided by the culturing base of Yangtze River Fisheries Research Institute (Wuhan, China) were held in tanks with a volume of 480 L for each tank flowing well water of 26 L/min (6 fish each tank). Water quality parameters for the water environment were controlled to an appropriate extent by daily measuring corresponding values, including pH at 7.2 ± 0.4, nitrite nitrogen levels ˂0.07 mg/L, total ammonia nitrogen levels ≤0.72 mg/L, and dissolved oxygen levels at 6.4–7.3 mg/L. The water temperature at either 18 ± 0.5 °C or 24 ± 0.5 °C used to carry out the pharmacokinetic experiments was kept by aquarium heater and air conditioner. Before drug treatment, grass carp were acclimated for 14 days while feeding a drug-free formula feed containing 28.00% crude proteins, 7.06% crude fat, 8.75% moisture, 15.00% crude fiber, and 15.63% ash [31] produced by the Nutritional Research Group in Yangtze River Fisheries Research Institute, Chinese Academy of Fishery Sciences, Wuhan, China. The blank plasma and tissues (liver, kidney, muscle + skin, and gill) used as negative control were collected from 10 fish without drug treatment and stored at −20 °C. All the experimental protocols were approved by the Fish Ethics Committee of Yangtze River Fisheries Research Institute, Chinese Academy of Fishery Sciences, Wuhan, China.

### 2.3. Experimental Design

The SDZ solution given to grass carp via oral gavage was made as a final concentration of 40 mg/mL by dissolving SDZ powder in pure water. In order to promote the dissovling of SDZ powder in water, a proper volume of sodium hydroxide solution (1 mol/L) was added to the SDZ solution (*v:v* = 1:10) to make sure that SDZ was completely soluble at 40 mg/mL in water. Fish were randomly divided into two treatment groups consisting of 66 fish in each group for two different temperatures of 18 and 24 °C. These temperatures were selected based on the local climate (i.e., 24 °C is approximately the average temperature in the summer and spring; and 18 °C is around the average temperature in the autumn and winter in Wuhan, China).

Before drug administration, the fish were fasted for 24 h. Then, the fish were weighed and administered with SDZ at the dose of 50 mg/kg in a liquid form by inserting a plastic tube attached to a 1-mL microinjector into the intestine. After oral administration, the treated fish were put into a separate tank for observing possible regurgitation of SDZ from the fish intestine. If the SDZ solution was regurgitated, the fish was removed from the study and replaced.

Subsequently, the samples of blood and tissues including liver, kidney, muscle + skin, and gill were collected from six fish at each time point of 0.083, 0.167, 0.5, 1, 2, 8, 16, 24, 48, 72, and 96 h after oral dosing. Approximately 2.0 mL of blood was drawn from the caudal vessels of each fish using a 2.5 mL-heparinized syringe with a 22 G needle. Afterward, the tissues of the liver, kidney, muscle + skin, and gill were also collected from each fish. After centrifugation of blood sample at 1500× *g* for 5 min at 4 °C, the plasma was pipetted into a new tube. All samples were stored at −20 °C until analysis. 

### 2.4. Sample Preparation and Analysis

The processing method of the sample is in line with the reported procedure by Wang, Luo, Xiao, Zhang, Deng, Tan and Jiang [16] with some modifications. In brief, the plasma and homogenized tissue samples (muscle + skin, liver, kidney, and gill) were thawed at 25 °C. One mL of plasma or 1 g of tissues was transferred into 10-mL polypropylene tubes. Five milliliters of ethyl acetate was blended with each sample, and then 0.5 g of anhydrous magnesium was weighed and added into each tube. The mixed samples were shaken for 2 min and then centrifuged for 5 min at 5000× *g* at 4 °C. The extracted supernatant was pipetted into another new tube. The remaining matrix was extracted again by another 5 mL of ethyl acetate using the same procedure. The resulting upper layer was merged with the former tube. Thereafter, the extracts were evaporated using a gentle nitrogen stream at 40 °C until completely dry. The dry residues were re-dissolved by 1 mL of 20% acetonitrile in water (0.1% formic acid). One milliliter of n-heptane was shaken with the solution, and then the mixture was centrifugated at 5000× *g* for 1 min. The upper layer was discarded, and the lower layer was filtered through 0.22-µm nylon filters. A 10-µL sample was used for ultra-performance liquid chromatography (UPLC) analysis.

A Waters ACQUITY^TM^ UPLC (Milford, MA, USA) coupled with a sampler manager with an autosampler, a binary solvent manager with a binary solvent pump, and an ultraviolet detector was employed to determine the concentrations of SDZ and ACT-SDZ in all collected samples. An ACQUITY UPLC BEH C18 column (1.7 μm, 2.1 × 100 mm) was used to elute chemicals at 30 °C. The detection wavelength was set as 280 nm. The mobile phase comprised pure water (0.1% formic acid) and acetonitrile with a proportion of 80:20 at 0.3 mL/min.

### 2.5. Calibration Curves and Recovery Rates

The blank plasma samples from grass carp without feeding target chemicals were fortified with a reference standard solution of SDZ and ACT-SDZ at final concentrations of 10, 20, 100, 500, 2000, 5000, and 20,000 μg/L for each chemical. Tissue samples (muscle + skin, liver, kidney, and gill) from untreated grass carp were also spiked with a standard solution of SDZ and ACT-SDZ to yield concentrations of 10, 20, 100, 500, 2000, 5000, and 20,000 μg/kg for each chemical. Samples were processed in the light of the method as described above, and each concentration was set as three parallels. Precision and accuracy were determined by analyzing five replicates of spiked plasma and tissue samples at 10, 100, and 5000 μg SDZ/L or /kg.

### 2.6. Pharmacokinetic Analysis

The mean concentrations of SDZ and ACT-SDZ in plasma and individual tissues at each sampling time point were calculated to derive the mean concentration versus time profiles for plasma and individual tissues for each temperature group. Next, a classical one-compartmental approach was used to analyze the SDZ and ACT-SDZ plasma and tissue mean concentration versus time profiles using Phoenix Winnonlin 7.0 (Certara, Inc., Princeton, NJ, USA). The following pharmacokinetic parameters were calculated: AUC_0–∞_ (area under concentration-time curve from 0 h to ∞), K01 (absorption rate constant), K10 (elimination rate constant), K01_HL (absorption half-life), K10_HL (elimination half-life), T_max_ (the time to reach the peak concentration), C_max_ (the peak concentration), V_F (the apparent volume of distribution per fraction of dose absorbed), and Cl_F (the apparent systemic total body clearance per fraction of dose absorbed). The apparent metabolic rate (AMR) for ACT-SDZ (i.e., relative exposure to ACT-SDZ out of total exposure to the combination of SDZ and ACT-SDZ) was calculated based on the method reported by [26,32] using the following equation.
$$\mathrm{AMR}=\frac{{\mathrm{AUC}}_{\mathrm{ACT}-\mathrm{SDZ}}}{{\mathrm{AUC}}_{\mathrm{ACT}-\mathrm{SDZ}}+{\mathrm{AUC}}_{\mathrm{SDZ}}}\;\times 100\%$$

## 3. Results

### 3.1. Method Validation

In this study, the analytical method had a limit of detection and limit of quantitation of 7 and 10 μg/L (or μg/kg), respectively, for both SDZ and ACT-SDZ in both plasma and tissues. The matrix-fortified calibration curves were established through a linear regression peak area with corresponding concentration, which exhibited good linearity by the coefficient of correlation R^2^ = 0.998 (the variations of concentration values used for establishing the calibration curve were all less than 15%). For samples with concentrations of SDZ and ACT-SDZ in plasma and tissues over the upper limit of quantification in the initial measurement, the remaining samples were diluted with corresponding blank plasma or tissue samples, and the measurement was repeated. The mean recovery rates of SDZ and ACT-SDZ were 83.5–92.1% in plasma and tissues. Their relative standard deviations for inter-day and intra-day precision were ≤10%.

### 3.2. Pharmacokinetic Characteristics of SDZ at Two Different Temperatures 

The concentrations of SDZ and ACT-SDZ in plasma and tissues of grass carp after single oral dosing at 50 mg/kg at two different temperatures are respectively listed in Table 1 and Table 2. The concentration vs. time profiles are shown in Figure 1 for 18 °C and Figure 2 for 24 °C. At the water temperature of 18 °C, the results showed that SDZ’s concentration reached a maximum value at 8 h in plasma and liver, and at 16 h in the kidney, muscle + skin, and gill, respectively. ACT-SDZ’s concentration reached the peak concentration at 8 h in liver, kidney, and gill, and at 48 h in plasma and muscle + skin, respectively. After that, the concentration of SDZ and its metabolite ACT-SDZ started to decrease gradually. At the water temperature of 24 °C, the peak concentration of SDZ in kidney and gill was observed earlier at 4 h after oral dosing, but at 8 h in plasma, liver, and muscle + skin. The time to peak concentration of ACT-SDZ was all at 8 h. In a comparison of drug concentration in plasma and tissues at 18 °C with 24 °C, the higher concentration was maintained at 18 °C after 16 h. Moreover, the concentrations of SDZ and its metabolite ACT-SDZ at the last sampling point of 96 h at 18 °C were much higher than those at 24 °C for plasma and all tissues. Additionally, an interesting phenomenon was found that the ACT-SDZ’s concentration was more than its parent drug in major metabolic/excretory organs (liver and kidney) regardless of 18 °C or 24 °C.

The calculated pharmacokinetic parameters in plasma and tissues are displayed in Table 3 for SDZ and Table 4 for ACT-SDZ. When comparing the results at 18 °C with those at 24 °C, the AUC of SDZ was considerably decreased by 50.00%, 53.75%, 61.70%, 70.93%, and 77.74% in gill, kidney, liver, muscle + skin, and plasma, respectively. There was a 15.52−88.03% decrease in the K01-HL and a corresponding increase in the K01 for gill, kidney, and plasma, but the increasing trends were exhibited in the liver (0.90 h vs.1.37 h) and muscle+skin (3.69 h vs. 8.68 h). The K10_HL also presented a distinctly declined tendency in plasma and tissues. The V_F was increased by about two-fold, and the total body clearance was raised by around five-fold. Regarding its metabolite ACT-SDZ, the consistent trends as the parent drug were presented in AUCs of gill, kidney, liver, muscle + skin, and plasma with notable reduction by 93.69%, 85.84%, 37.95%, 97.92%, and 95.38% by comparing values at 18 °C to 24 °C. The K01_HL was reduced in gill, kidney, muscle + skin, and plasma except liver with a slight rise (1.4 h vs. 1.6 h). The K10_HL was decreased as well by 20.22%, 72.14%, 47.79%, 98.76%, and 41.27% in gill, kidney, liver, muscle+skin, and plasma, respectively. The AMRs of ACT-SDZ in plasma and tissues were also calculated using a published method [26], but those values might be overestimated. Possible reasons are discussed in the Discussion section below.

## 4. Discussion

The present study examined the pharmacokinetic properties of SDZ and its metabolite, ACT-SDZ, following oral gavage at a single dose of 50 mg/kg at different temperatures. ACT-SDZ is one of the metabolites generated from the parent drug SDZ in animals and humans. In general, sulfonamides are metabolized by oxidation, acetylation, and glucuronidation reactions, with N_4_-acetylation and N_4_-glucuronidation being the main metabolic pathways [33,34,35]. However, there are metabolic divergences of sulfonamides in different animal species. For example, N_4_-glucuronide metabolite is produced in humans, but not in pigs [34]. In fish, N_4_-acetylation has been proved to be the major metabolic pathway, with the N_4_-acetylation metabolite presenting a high concentration in the plasma and tissues [36,37,38,39,40]. In the present study, the N_4_-acetylation metabolite ACT-SDZ was extensively generated in grass carp whether at low or high temperature, especially in the liver (at 18 and 24 °C), kidney (at 18 and 24 °C), and gill (at 18 °C) with a higher concentration than the parent drug, suggesting the N_4_-acetylation metabolite was mainly produced in the liver and excreted from the kidney. The resulting elimination half-lives (K10_HL) of ACT-SDZ were longer than that of SDZ in plasma and tissues, with a possible reason that ACT-SDZ was converted to the parent drug via a deacetylation process [40]. In addition, the low clearance of ACT-SDZ may also enlarge its K10_HL. These findings were consistent with the results reported by Uno, Aoki and Ueno [40] in rainbow trout (*Oncorhynchus mykiss*) given sulfadimethoxine at a dose of 200 mg/kg via oral administration at 15 °C. On contrary, N_4_-acetyl sulfadimethoxine concentration was significantly higher at 20 °C than at 10 °C in carp (*Cyprinus carpio L*.), whereas there was no significant difference in trout (*Salmo gairdneri Richardson*) [41]. These differences may be possibly due to disparate metabolic capacity existing in different fish species, resulting in metabolite formation divergence.

The results demonstrated that the shifts in the environment temperature may have prominent effects on the pharmacokinetic characteristics of SDZ and ACT-SDZ in grass carp. At 18 °C, the absorption half-lives (K01_HL) of SDZ and ACT-SDZ were longer than at 24 °C in gill, kidney, and plasma except in liver and muscle + skin, and corresponding elimination half-lives (K10_HL) were longer in all tissues and plasma. These differences may be primarily due to a higher temperature altering the fish’s physiological parameters, including shortening blood circulation time, enlarging cardiac output, and enhancing organs’ metabolic rates thereby causing the shifts in K01_HL and K10_HL [42,43,44]. It has been reported that a temperature increase of 1 °C corresponded to a 10% increase in metabolic and excretory in fish [45], indicating that the temperature is a predominant influential factor on drug absorption, distribution, metabolism, and excretion in poikilotherms. Therefore, the development of drugs used in aquatic animals must consider the effect of temperature on drug pharmacokinetics. In the pharmacokinetics of SDZ in other fish, the authors did not study the difference in pharmacokinetics at different temperatures. It is reported that K01_HL and K10_HL of SDZ were 5.70 and 25.90 h in plasma of mandarin fish (*Siniperca chuatsi*) after a single oral administration of SDZ and trimethoprim (TMP) at 120 mg/kg (SDZ:TMP = 5:1) at 28 °C [16]. The K01_HL value was longer than that in the present study, whether at 18 or 24 °C, but the K10_HL value was longer than that at 24 °C but shorter than at 18 °C in this study. In another study in Atlantic salmon (*Salmo salar*) at 10 °C after a single oral dose of SDZ/TMP at 25.0/5.0 mg/kg, the K10_HL of SDZ was 27.00 h in plasma that was longer than that in grass carp at 24 °C but less than that at 18 °C. These differences may be caused by disparate dosages, water temperatures, and fish species, as well as drug-drug interaction. In addition, many scientists have conducted a pharmacokinetic study of other drug classes in fish at different temperatures, and consistent findings with this study have also been reported. Rairat, Hsieh, Thongpiam, Sung and Chou [24] found that the absorption half-lives were decreased from 0.59 h to 0.3 h, and the elimination half-lives were also reduced from 12.49 h to 7.9 h following FF oral treatment at a dose of 15 mg/kg in Nile tilapia (*Oreochromis niloticus*) at three different temperatures from 24 °C to 32 °C. Yang, Yang, Wang, Kong and Liu [26] reported that K10_HF of FF was reduced from 22.86 h to 9.56 h, and K10_HF of its metabolite, florfenicol amine (FFA), was declined from 27.79 h to 15.44 h as temperature increased from 10 °C to 25 °C after single oral dosing of FF at 10 mg/kg in crucian crap (*Carassius auratus*). In a pharmacokinetic study of oxytetracycline in sea bass (*Dicentrarchus labrax*) following an intravascular administration at 40 mg/kg, the K01_HL at 13.5 °C was about five-fold of that at 22 °C, and K10_HF at 13.5 °C was more than seven-fold of that at 22 °C [30]. Similar phenomena were found in a study of oxolinic acid and flumequine in fish [27,28]. Therefore, the temperature-dependent effect of K01_HL and K10_HL appears to be common for the pharmacokinetics of drugs in fish.

The T_max_ of SDZ at 18 °C were larger than those at 24 °C by about ~2–4 fold in gill, kidney, and plasma, except in liver (5.34 h versus 5.78 h) and muscle+skin (14.5 h versus 15.23 h) being comparable at the two temperatures. The C_max_ at 18 °C in muscle+skin and plasma were higher than at 24 °C, but that in gill was less than at 24 °C and its value in kidney and liver were comparable to at 24 °C. The T_max_ in ACT-SDZ also presented a decreasing trend in gill, kidney, liver, and plasma from 18 °C to 24 °C. Generally, the C_max_ at 18 °C was higher than at 24 °C in gill, kidney, and plasma, except in the liver. From these results, the T_max_ and C_max_ of SDZ and ACT-SDZ displayed the same trend from 18 °C to 24 °C primarily because of a shorter K01_HL existing at a higher temperature. In another pharmacokinetic study of a structurally-similar drug sulfadimethoxine, T_max_ values at 10 °C were much longer than at 20 °C in carp and trout, but the C_max_ at 10 °C was only slightly less than that at 20 °C after intravenous administration at a dose of 100 mg/kg under different temperatures [41]. Yang, Yang, Wang, Kong and Liu [26] reported that T_max_ of FF and FFA at 25 °C were respectively shortened to 0.86 h and 3.63 h from that at 10 °C, and the C_max_ of FF and FFA were respectively increased by 1.21 mg/L and 0.25 mg/L. In these studies, the trend in T_max_ at different temperatures is similar to this study, but the tendency in C_max_ from low temperature to high temperature is different from the present study partly due to disparate drug administration, fish species, oral dosage and drug properties.

Our results showed that the AUC values of both SDZ and ACT-SDZ in plasma and tissues were considerably decreased from 18 °C to 24 °C. This is possibly owing to the increased temperature accelerating drug absorption and depletion. Wang, Luo, Xiao, Zhang, Deng, Tan and Jiang [16] reported the AUC of SDZ in plasma of mandarin fish is 1601.60 µg * h/mL after a single oral dose of SDZ and TMP at 120 mg/kg (SDZ:TMP = 5:1) at 28 °C, which is larger than that at 24 °C in grass carp but less than at 18 °C in this study. The smallest AUC was presented in Atlantic salmon, which may be due to the smallest given dose in this species [18]. The V_F and the Cl_F were correspondingly increased from 18 °C to 24 °C, which is consistent with previous findings in Nile tilapia (*Oreochromis niloticus*) by administering a single dose of FF [24]. On the contrary, the results of doxycycline pharmacokinetics in grass carp exhibited a higher value of V_F and Cl_F at a lower temperature (18 °C vs. 24 °C) partly because different drugs possess disparate metabolic profiles [22].

In this study, the AMR of ACT-SDZ was calculated by the method of AUC [26,32,46]. The results showed that the estimated AMRs were influenced by the increase of temperature. Its values at 18 °C were markedly higher than at 24 °C in gill (62.8% vs. 17.57%), kidney (94.30% vs. 83.52%), muscle+skin (79.07% vs. 21.28%), and plasma (30.12% versus 9.10%) except in liver (65.60% vs. 75.55%). This trend was opposite to the previous study on FF with a high AMR of 40.23% in plasma at 20 °C, but a relatively low AMR of 30.91% at 10°C [26]. The authors thought that these different results might be due to the sampling time points. In the present study, the absorption of ACT-SDZ was integrally included in the sampling time points, whereas we only collected 2–3 time points during the elimination phase until 96 h. Therefore, the AUCs in plasma and diverse tissues might be overestimated or underestimated because the extrapolated AUCs after 96 h were completely dependent on the simulation of the supposed compartmental model.

An imperative role of pharmacokinetic studies is to establish or adjust the dosage regimen based on pharmacokinetic results in combination with antibacterial experimental results obtained in vivo or in vitro. It is reported that the minimum inhibitory concentration (MIC) 50 and MIC 90 of SDZ were respectively 3.2 and 6.4 µg/mL, 1.6 and l.6 µg/mL, 1.0 and 1.6 µg/mL, and 1.6 and 3.2 µg/mL for *Aeromonas salmonicida, Vibrio anguillarum, V. salmonicida* and *Yersinia ruckeri* [18]. In this study, the concentration of SDZ was more than 6.4 µg/mL from 0.5 to 96 h in plasma at 18 °C, and from 0.167 to 48 h in plasma at 24 °C. Note that the present study measured the total concentration of SDZ in plasma, but the free drug concentration in plasma should be used to compare with MIC when designing optimal therapeutic regimens [47,48]. The free fraction of SDZ in grass carp has not been reported, but the free fraction of SDZ in plasma of humans is around 52 to 62% [49]. Assuming the free fraction of SDZ in plasma of grass carp is similar to humans, the concentration of free SDZ would be more than 6.4 µg/mL from 1 to 96 h in plasma at 18 °C, and from 1 to 24 h in plasma at 24 °C. To guarantee enough concentrations of free SDZ in plasma, an oral dose of SDZ at 50 mg/kg should be given once per 96 h at 18 °C and once per 24 h at 24 °C. Additional studies are needed to determine the plasma protein binding percentage of SDZ in grass carp.

Another contribution of this study is to provide necessary chemical-specific pharmacokinetic parameters and extensive original pharmacokinetic and tissue distribution data that are essential to build a physiologically based pharmacokinetic (PBPK) or a population pharmacokinetic (PopPK) model for SDZ in grass carp. PBPK and PopPK models are an important tool in drug discovery and development, dose optimization, human health risk assessment, tissue residue, and withdrawal interval estimation [50,51,52,53,54]. This is a direction for future studies.

The present study has some limitations. First, we used the force-administering method to give SDZ to fish. Although this method can ensure that we give a consistent dose in unit of mg/kg to all fish, it should be noted that it is not the method that is commonly used in the actual aquaculture practice, which is via medication feed. The bioavailability may be different between the force-administering gavage method and the medicated feed method. However, the present study chose the oral gavage method, rather than the medication feed method because of the following issues that are associated with medication feed administration method: (1) it is difficult to guarantee fish will eat out all medication feed; (2) the given dose is not consistent between fish due to the difference in feed intake; (3) the feed may influence the pharmacokinetic parameters; and (4) the pharmacokinetic characteristics may not be identified due to the huge interindividual variability in the oral dose via medication feed. To better understand the pharmacokinetics of SDZ in actual aquaculture, future studies should consider administering drug to fish using both oral gavage and medicated feed methods, and then compare the differences in the pharmacokinetic results. In addition, the present study used a classical one-compartmental approach to analyze the mean concentration versus time profiles of SDZ and ACT-SDZ in plasma and individual tissues for each temperature group. As such, the variability of the calculated pharmacokinetic parameters was not characterized. Future studies may consider using more advanced pharmacokinetic approaches, such as PopPK modeling via nonlinear mixed effects approach [54,55,56] and population PBPK modeling via Bayesian analysis with Markov chain Monte Carlo simulation [57,58] to characterize the variability of pharmacokinetic parameters of SDZ and ACT-SDZ in grass carp.

## 5. Conclusions

The pharmacokinetics of SDZ and its main metabolite ACT-SDZ were investigated in grass carp (*Ctenopharyngodon idella*) following a single oral dose of 50 mg/kg at 18 and 24 °C. The results showed that changes in water temperature notably affect the pharmacokinetic properties and tissue disposition of SDZ and ACT-SDZ in grass carp. As the temperature increased from 18 to 24 °C, K10_HF, C_max_, and AUC_0–∞_ of SDZ were decreased in plasma and tissues, and Cl_F and V_F were increased in plasma. The metabolite of ACT-SDZ was generated extensively and widely distributed to plasma and tissues, especially in the liver, kidney, and gill. The pharmacokinetic characteristics of ACT-SDZ were similar to that of the parent drug. Our study further supports that it is imperative to design different therapeutic regimens of SDZ at different temperatures for disparate pathogens to avoid treatment failure and risk of drug resistance. Based on our results and considering reported MIC values for different bacteria, we recommend an oral dose of SDZ at 50 mg/kg once per 96 h at 18 °C and once per 24 h at 24 °C to treat bacterial diseases against *A. salmonicida*, *V. anguillarum*, *V. salmonicida*, and *Y. ruckeri*.

## Figures and Tables

**Figure 1 pharmaceutics-14-00712-f001:**
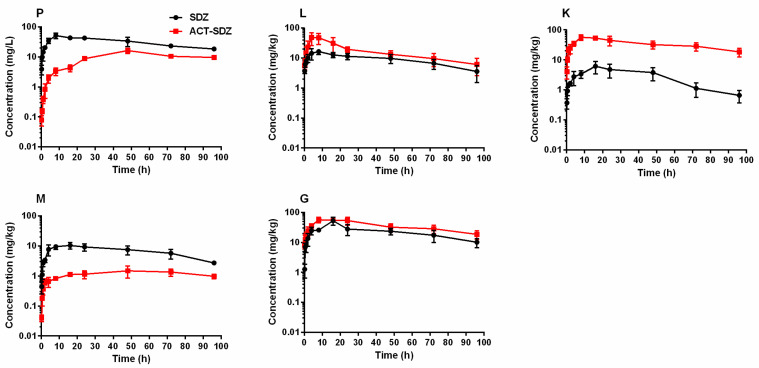
Semi-logarithmic concentration-time profiles of sulfadiazine and its main metabolite N-acetyl sulfadiazine in plasma (**P**), liver (**L**), kidney (**K**), muscle+skin (**M**), and gill (**G**) of grass carp (*Ctenopharyngodon idella*) following a single oral dose of sulfadiazine at 50 mg/kg at 18 °C. Abbreviations on the figure: SDZ, sulfadiazine; ACT-SDZ, N-acetyl sulfadiazine.

**Figure 2 pharmaceutics-14-00712-f002:**
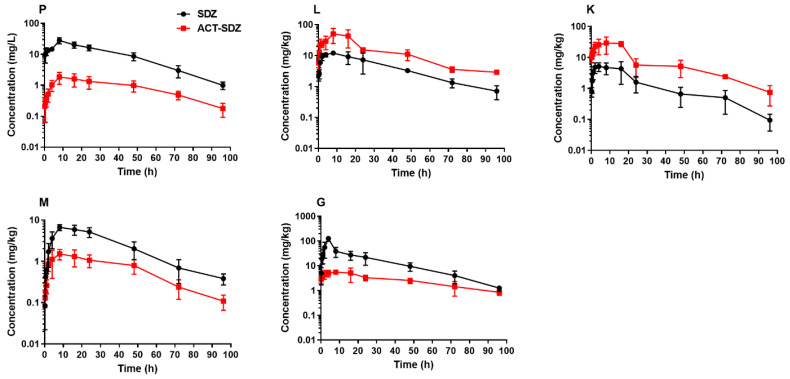
Semi-logarithmic concentration-time profiles of sulfadiazine and N-acetyl sulfadiazine in plasma (**P**), liver (**L**), kidney (**K**), muscle + skin (**M**), and gill (**G**) of grass carp (*Ctenopharyngodon idella*) following a single oral dose of sulfadiazine at 50 mg/kg at 24 °C. Abbreviations on the figure: SDZ, sulfadiazine; ACT-SDZ, N-acetyl sulfadiazine.

**Table 1 pharmaceutics-14-00712-t001:** Sulfadiazine and N-acetyl sulfadiazine concentrations in plasma and tissues of grass carp (*Ctenopharyngodon idella*) after oral administration of a single dose of sulfadiazine at 50 mg/kg at 18 °C.

Time (h)	Concentration (mg/L or mg/kg)									
	Plasma		Liver		Kidney		Muscle + Skin		Gill	
	SDZ	ACT-SDZ	SDZ	ACT-SDZ	SDZ	ACT-SDZ	SDZ	ACT-SDZ	SDZ	ACT-SDZ
0.167	3.92 ± 1.38	0.08 ± 0.03	3.71 ± 0.68	5.56 ± 1.52	0.37 ± 0.14	4.04 ± 1.77	0.44 ± 0.19	0.04 ± 0.01	1.26 ± 0.61	8.04 ± 1.83
0.5	9.92 ± 1.15	0.16 ± 0.03	6.17 ± 1.24	7.43 ± 2.11	0.92 ± 0.28	10.49 ± 2.10	1.11 ± 0.35	0.18 ± 0.08	6.99 ± 2.34	12.49 ± 4.66
1	14.44 ± 7.18	0.38 ± 0.10	7.97 ± 1.36	14.94 ± 8.40	1.47 ± 0.10	20.47 ± 5.89	2.71 ± 0.66	0.37 ± 0.14	8.06 ± 3.50	16.47 ± 5.46
2	20.29 ± 1.49	0.84 ± 0.43	10.05 ± 2.90	22.98 ± 13.62	1.66 ± 0.32	24.38 ± 8.42	3.39 ± 0.59	0.65 ± 0.15	13.58 ± 6.11	24.38 ± 8.42
4	34.73 ± 7.17	1.94 ± 0.60	14.42 ± 5.97	48.55 ± 22.22	2.75 ± 1.35	35.49 ± 7.04	7.72 ± 3.08	0.66 ± 0.24	24.85 ± 7.03	35.49 ± 7.04
8	51.36 ± 10.79	3.36 ± 0.96	16.37 ± 3.23	47.62 ± 19.19	3.37 ± 0.94	57.1 ± 11.78	9.48 ± 1.69	0.83 ± 0.12	25.80 ± 1.58	57.10 ± 11.78
16	43.19 ± 4.87	4.36 ± 1.13	12.76 ± 2.50	31.37 ± 17.22	6.16 ± 2.71	53.54 ± 5.34	10.44 ± 2.46	1.12 ± 0.08	52.85 ± 15.51	54.90 ± 5.34
24	42.61 ± 6.67	8.73 ± 1.40	11.42 ± 2.62	19.53 ± 4.27	4.77 ± 2.31	44.90 ± 16.05	9.31 ± 2.51	1.15 ± 0.34	27.86 ± 11.05	53.54 ± 16.05
48	33.81 ± 11.79	16.26 ± 3.26	9.70 ± 3.19	13.36 ± 4.17	3.79 ± 1.77	32.26 ± 9.44	7.53 ± 2.48	1.49 ± 0.64	23.49 ± 5.89	32.26 ± 9.44
72	23.10 ± 1.10	10.53 ± 1.23	6.67 ± 2.44	9.58 ± 4.48	1.13 ± 0.56	28.56 ± 8.71	5.75 ± 2.07	1.35 ± 0.37	17.36 ± 7.48	28.56 ± 8.71
96	18.42 ± 2.70	9.57 ± 1.08	3.56 ± 2.03	6.11 ± 3.54	0.66 ± 0.29	18.50 ± 5.84	2.72 ± 0.19	0.98 ± 0.17	10.09 ± 3.37	18.50 ± 5.84

Note: SDZ, sulfadiazine; ACT-SDZ, N-acetyl sulfadiazine.

**Table 2 pharmaceutics-14-00712-t002:** Sulfadiazine and N-acetyl sulfadiazine concentrations in plasma and tissues of grass carp (*Ctenopharyngodon idella*) after oral administration of a single dose of sulfadiazine at 50 mg/kg at 24 °C.

Time (h)	Concentration (mg/L or mg/kg)									
	Plasma		Liver		Kidney		Muscle + Skin		Gill	
	SDZ	ACT-SDZ	SDZ	ACT-SDZ	SDZ	ACT-SDZ	SDZ	ACT-SDZ	SDZ	ACT-SDZ
0.167	9.46 ± 1.22	0.21 ± 0.15	2.32 ± 0.82	5.74 ± 2.89	0.80 ± 0.27	10.02 ± 3.36	0.08 ± 0.06	0.13 ± 0.05	5.08 ± 3.39	2.90 ± 0.93
0.5	10.23 ± 5.01	0.27 ± 0.21	2.61 ± 0.85	12.72 ± 8.69	1.74 ± 1.04	13.89 ± 4.41	0.42 ± 0.16	0.18 ± 0.06	19.51 ± 7.86	3.76 ± 0.70
1	12.71 ± 3.54	0.35 ± 0.15	5.93 ± 0.57	21.93 ± 12.21	3.50 ± 1.59	15.56 ± 4.48	0.60 ± 0.10	0.26 ± 0.10	28.56 ± 16.45	4.02 ± 1.17
2	13.35 ± 1.78	0.52 ± 0.24	9.33 ± 2.37	24.09 ± 5.68	4.86 ± 1.82	22.76 ± 8.07	1.72 ± 1.00	0.85 ± 0.29	55.18 ± 34.74	4.68 ± 1.83
4	14.78 ± 2.07	1.03 ± 0.40	10.86 ± 1.82	28.91 ± 13.37	5.14 ± 1.62	25.54 ± 13.34	3.59 ± 1.58	1.12 ± 0.74	127.13 ± 28.75	5.06 ± 1.50
8	28.12 ± 6.33	1.85 ± 0.77	12.10 ± 0.62	50.13 ± 25.48	4.71 ± 1.94	29.37 ± 16.67	6.65 ± 1.07	1.52 ± 0.43	38.74 ± 16.91	5.56 ± 0.72
16	20.33 ± 4.55	1.60 ± 0.73	9.24 ± 4.04	42.69 ± 25.16	4.35 ± 2.98	28.07 ± 5.42	5.90 ± 1.59	1.31 ± 0.58	27.62 ± 10.71	5.12 ± 2.99
24	16.47 ± 3.44	1.33 ± 0.58	7.25 ± 4.73	15.32 ± 2.99	1.59 ± 0.87	5.67 ± 3.44	5.17 ± 1.38	1.07 ± 0.37	22.04 ± 11.5	3.34 ± 0.87
48	8.67 ± 2.50	0.99 ± 0.38	3.27 ± 0.43	11.07 ± 4.19	0.67 ± 0.42	5.17 ± 2.93	2.03 ± 0.92	0.80 ± 0.31	9.55 ± 3.57	2.56 ± 0.66
72	3.01 ± 1.28	0.49 ± 0.15	1.33 ± 0.43	3.60 ± 0.71	0.50 ± 0.36	2.40 ± 0.18	0.69 ± 0.41	0.24 ± 0.12	4.04 ± 2.14	1.45 ± 0.86
96	1.00 ± 0.27	0.18 ± 0.08	0.71 ± 0.34	2.89 ± 0.41	0.09 ± 0.05	0.75 ± 0.48	0.38 ± 0.11	0.11 ± 0.04	1.26 ± 0.32	0.85 ± 0.20

Note: SDZ, sulfadiazine; ACT-SDZ, N-acetyl sulfadiazine.

**Table 3 pharmaceutics-14-00712-t003:** Pharmacokinetic parameters of sulfadiazine in gill, kidney, liver, muscle + skin and plasma of grass carp (*Ctenopharyngodon idella*) after a single oral dose of 50 mg/kg at 18 and 24 °C.

Parameters	Unit	Gill		Kidney		Liver		Muscle + Skin		Plasma	
		18 °C	24 °C	18 °C	24 °C	18 °C	24 °C	18 °C	24 °C	18 °C	24 °C
AUC_0–∞_	h×mg/L	2873.5	1436.6	294.8	136.3	1159.7	444.2	864.8	251.4	4612.6	1026.7
K01_HL	h	3.63	1.04	7.77	0.93	0.90	1.37	3.69	8.68	2.32	1.96
K10_HL	h	44.15	13.75	20.68	14.69	50.25	21.06	44.99	13.02	59.36	23.35
K01	1/h	0.19	0.66	0.09	0.74	0.77	0.51	0.19	0.08	0.3	0.35
K10	1/h	0.02	0.05	0.03	0.05	0.01	0.03	0.02	0.05	0.01	0.03
T_max_	h	14.25	4.20	17.58	3.96	5.34	5.78	14.50	15.23	11.29	7.66
C_max_	mg/L	36.07	58.61	5.48	5.34	14.86	12.09	10.66	5.95	47.21	24.28
V_F	L/kg	NA	NA	NA	NA	NA	NA	NA	NA	0.932	1.641
Cl_F	L/h/kg	NA	NA	NA	NA	NA	NA	NA	NA	0.0109	0.0487

Note: AUC_0–∞_, the area under concentration–time curve from 0 h to ∞; K01_HL, the absorption half-life; K10_HL, the elimination half-life; K01, the absorption rate constant; K10, the elimination rate constant; T_max_, the time to reach the peak concentration; C_max_, the peak concentration; V_F, the apparent volume of distribution per fraction of dose absorbed; Cl_F, the apparent systemic total body clearance per fraction of dose absorbed; NA, not available or not applicable.

**Table 4 pharmaceutics-14-00712-t004:** Pharmacokinetic parameters of N-acetyl sulfadiazine in gill, kidney, liver, muscle + skin and plasma of grass carp (*Ctenopharyngodon idella*) after orally administered sulfadiazine at a dose of 50 mg/kg at 18 and 24 °C.

Parameters	Unit	Gill		Kidney		Liver		Muscle + Skin		Plasma	
		18 °C	24 °C	18 °C	24 °C	18 °C	24 °C	18 °C	24 °C	18 °C	24 °C
AUC_0–∞_	h×mg/L	4851.9	306.3	4877.9	690.9	2211.9	1372.4	3266.4	68.0	1987.9	91.9
K01_HL	h	2.59	0.20	2.22	0.67	1.44	1.55	2.41	3.43	44.35	4.29
K10_HL	h	51.29	40.92	56.25	15.67	36.91	19.27	1836.27	22.81	44.75	26.28
K01	1/h	0.27	3.53	0.31	1.03	0.48	0.45	0.29	0.20	0.02	0.14
K10	1/h	0.01	0.02	0.01	0.04	0.02	0.04	0.00038	0.03	0.02	0.03
T_max_	h	11.74	1.52	10.77	3.20	7.00	6.12	49.47	10.85	64.27	13.41
C_max_	mg/L	55.95	5.06	52.64	26.53	36.42	39.61	1.15	1.49	11.38	1.70
V_F	L/kg	NA	NA	NA	NA	NA	NA	NA	NA	NA	NA
Cl_F	L/h/kg	NA	NA	NA	NA	NA	NA	NA	NA	NA	NA
AMR	%	62.80	17.57	94.30	83.52	65.60	75.55	79.07	21.28	30.12	9.10

Note: AUC_0–∞_, the area under concentration–time curve from 0 h to ∞; K01_HL, the absorption half-life; K10_HL, the elimination half-life; K01, the absorption rate constant; K10, the elimination rate constant; T_max_, the time to reach the peak concentration; C_max_, the peak concentration; V_F, the apparent volume of distribution per fraction of dose absorbed; Cl_F, the apparent systemic total body clearance per fraction of dose absorbed; NA, not available or not applicable; AMR, the apparent metabolic rate of N-acetyl sulfadiazine.

## Data Availability

Not applicable.

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
