# Peer review of "Comparative Pharmacokinetics of Sulfadiazine and Its Metabolite N4-Acetyl Sulfadiazine in Grass Carp (Ctenopharyngodon idella) at Different Temperatures after Oral Administration"

_pharmaceutics, 2022, doi:10.3390/pharmaceutics14040712_

Round 1

Reviewer 1 Report

Given the high marginality and demand for grass carp (Ctenopharyngodon idella) by consumers, it is extremely important to study the pharmacokinetics and metabolism of drugs that are used in the cultivation of grass carp and can indirectly affect people who eat it.

Thus, the studies conducted by the authors of the article on the study of the plasma pharmacokinetics and tissue distribution of sulfadiazine (SDZ) and its main metabolite, N4-acetylsulfadiazine (ACT-SDZ) seem important and relevant.

Meanwhile, in the course of familiarization with the presented materials, I had a number of questions:

  1. I believe that the article should be supplemented with information and the main results of similar studies of the pharmacokinetics of sulfadiazine, previously conducted on other living objects, and also use these data when formulating the conclusions of this study. What are the common patterns and differences?
  2. Why were the temperatures 18 and 24 °C chosen?
  3. Has the study been conducted on a spawning population? Are sulfadiazine and its metabolites found in caviar? Are there any consequences of using the drug for grass carp fry?

Reviewer 2 Report

The manuscript is well structured and well-written. It provides understanding on Pharmacokinetic parameters of SDZ as a function of temperature effect. The results are systematically presented and well narrated. However, few formatting mistakes have been noticed that should be removed before further processing. I recommend it for publication after thoroughly going through to eliminate all formatting mistakes

What was the reason of using temperature range of 18C to 24C for evaluating the Pharmacokinetic parameters? Does it make sense to use wider temperature range? Like from 10C to 30C, If data has been produced using the same temperature range, How the outcome was different?

It was used 40 mg/L of the dose of SDZ, Does this does is a therapeutic dose? Also why 1 mole/L NaOH was added to the SDZ solution and what is meant by a proper volume (how much volume was added). Does it affect the final outcome? Did you use SDZ solution without NaOH addition? In first line it was mentioned 40 mg/L and then in the third line of Experimental design it was mentioned 40 mg/ml. Please check and correct which concentration was given to the fishes.

Reviewer 3 Report

This is an interesting and well-written article describing the pharmacokinetics of sulfadiazine in carp. I have three  major comments and a few minor comments.

Major comment

1-The authors used a force-feeding method to carry out their study. It is essential to know if this method of force-feeding is used routinely to treat carp or if it is only an experimental procedure?  if under field conditions, gavage is not used, then this point must be seriously discussed because there is no guarantee that the bioavailability will be the same by gavage and by spontaneous ingestion of an SDZ formulation.

2-The data analysis is not properly described and I suspect it was not conducted optimally. The authors say they used a compartmental approach and it is unclear whether this is a classic two-stages analysis method or a population modeling? POP PK  can be strongly recommended for this kind of toxicokinetic data set. Indeed these data set are often imbalanced and the NLMEM is the appropriate tool in this case (1) and more broadly, the NLME is recommended to analyse toxicokinetic data (2) ,  (3).  I observe that the authors did not report any statistical parameters (SD, SE) for their computed parameters such as the AUC: is it because these values were too high and would have led to not concluding differences for AUC obtained at the two different external temperature?  I strongly suggest to analyse these data with a NCA using the Phoenix sparse data option. The advantage of the sparse option is to  use the subject information to calculate standard errors that will account for any correlations in the data resulting from repeated sampling of individual animals. Standard error of the mean AUC will be calculated as described in Nedelman and Jia (1998), using a modification in Holder (2001), and will account for any correlations in the data resulting from repeated sampling of individual animals. (see Phoenix Doc; page 142) see also (4),   (5)

3-to discuss relevance of your plasma concentrations regarding MIC data, you have to take into account the extent of plasma protein binding (not reported). In addition, total tissular concentration has no meaning (6)

Minor comments

line 28 and 29: add apparent for Vd/F and Cl/F

line 167: the term apparent metabolic rate in not appropriate; it is rather a relative exposure

line 177 a high coefficient of correlation do not prove linearity

line 181 delete percentage or term that as a coefficient of variation (CV%)

line 192 and elsewhere: For Tmax, the authors must keep in mind that it is a hybrid parameter which depends as much on ka as on K01 and that the ka which is measured depends itself on the bioavailability. For these reasons, the authors must be more careful in their discussion of Tmax , a modification of Tmax does not necessarily mean a greater or lesser rate of absorption. Moreover for tissues, it is not a rate of absorption but rather an apparent rate of invasion

table 3: round up AUC values but give more figures for Cl/F

Line 245: round up

Line 265: explanation of  higher persistency of the metabolite cannot be only  some back deacetylation but rather , a lower clearance of the metabolite

Line 276 not a rate of absorption but of invasion

Line 290 FF is florfenicol?

References

  1. Schoemaker RC, Cohen AF. 1996. Estimating impossible curves using NONMEM. Br J Clin Pharmacol 42:283–290.
  2. Hing JP, Woolfrey SG, Greenslade D, Wright PM. 2001. Analysis of toxicokinetic data using NONMEM: impact of quantification limit and replacement strategies for censored data. J Pharmacokinet Pharmacodyn 28:465–479.
  3. Hing JP, Woolfrey SG, Greenslade D, Wright PM. 2001. Is mixed effects modeling or naïve pooled data analysis preferred for the interpretation of single sample per subject toxicokinetic data? J Pharmacokinet Pharmacodyn 28:193–210.
  4. Nedelman JR, Gibiansky E, Lau DT. 1995. Applying Bailer’s method for AUC confidence intervals to sparse sampling. Pharm Res 12:124–128.
  5. Nedelman JR, Gibiansky E. 1996. The variance of a better AUC estimator for sparse, destructive sampling in toxicokinetics. J Pharm Sci 85:884–886.
  6. Mouton JW, Theuretzbacher U, Craig WA, Tulkens PM, Derendorf H, Cars O. 2007. Tissue concentrations: do we ever learn? Journal of Antimicrobial Chemotherapy 61:235–237.
